# Synthesis and Characterization of New Layered Double Hydroxide-Polyolefin Film Nanocomposites with Special Optical Properties

**DOI:** 10.3390/ma12213580

**Published:** 2019-10-31

**Authors:** Fuensanta Monzó, Ana Vanessa Caparrós, Diego Pérez-Pérez, Alejandro Arribas, Ramón Pamies

**Affiliations:** 1Centro Tecnológico del Calzado y del Plástico, 30840 Alhama De Murcia, Spain; f.monzo@ctcalzado.org (F.M.); a.arribas@ctcalzado.org (A.A.); 2Servicio de Apoyo a la Investigación Tecnológica, Universidad Politécnica de Cartagena, 30202 Cartagena, Spain; ana.caparros@upct.es; 3Departamento de Ingeniería Mecánica, Materiales y Fabricación. Universidad Politécnica de Cartagena, 30202 Cartagena, Spain; diegoperezperez23@gmail.com

**Keywords:** LDH, low density polyethylene (LDPE), rheology, optical properties, films, composites

## Abstract

In this study, we have synthesized new double layered hydroxides to be incorporated to low density polyethylene thermoplastic matrix. These new composites present promising applications as materials to build greenhouses due to the enhancement of their optical properties. A characterization of the modified nanoclay has been performed by means of X-ray fluorescence (XRF), X-ray Diffraction (XRD), Thermogravimetric analysis (TGA), and Fourier-transform infrared spectroscopy (FTIR). We have prepared a series of polyolefin-based films to evaluate the effect of the addition of a whitening agent (disodium 2,2′-((1,1′-biphenyl)-4,4′-diyldivinylene)bis(benzenesulfonate)), the modified hydrotalcite-like material and a commercial dispersant. The rheological and mechanical characterization of the films have proved that the inclusion of the modified-layered double hydroxides (LDHs) do not substantially affect the processing and mechanical performance of the material. On the other hand, optical properties of the nanocomposites are improved by reducing the transmission in the UVA region.

## 1. Introduction

One of the most common types of degradation of polymers is induced by Ultraviolet (UV) light radiation which produces highly reactive free radicals. The so-called aging process is a result of photodegradation and color fading of polymers [1,2]. A wide range of additives are included in thermoplastic films to avoid UV light aging. These UV blocking compounds are classified based on the blocking mechanism: shielding by inorganic additives and UV light absorption by organic species. The thermal performance of the final product is one of the most important aspects to be taken into consideration since the processing of thermoplastic films occurs at high temperatures. In this case, inorganic additives usually are more stable at high temperatures than the organic additives. Besides, the UV energy absorbed can be dissipated as heat promoting a thermal aging of the thermoplastic matrix causing a loss of the mechanical properties [3,4]. In addition, organic UV absorbers migrate to the surface of the plastic due to their volatility and different polarity, so the plastic matrix loses its UV protection over time [5]. However, there are several strategies to reduce these negative effects. For example, the usage of layered double hydroxides (LDH) as host matrix of organic compounds results into a hybrid absorber which can block the UV light, increasing the thermal and photostability of the nanocomposite [6].

Layered double hydroxides (LDH) are anionic clays represented by the general formula of [M^2+^_1−x_M^3+^_x_(OH)_2_]^x+^(A^n−^)_x/n_·mH_2_O, where M^2+^ and M^3+^ are divalent and trivalent metallic cations and A^n−^ depicts the anions which can be present in the hydrated interlayer galleries of the clay [7,8,9,10,11]. The value of x is in the range of 0.20–0.33 [12]. Therefore, positively-charged metal hydroxide layers are intercalated with anions in order to keep the materials electroneutrality [13,14]. The cations fill octahedral sites having hydroxide ions in the vertices. These anions are weakly bound to the structure and ion exchange reactions are suitable for the modification of LDHs [15,16,17]. LDHs composed of Mg and Al hydroxyl sheets with carbonate ions as counter anions and M^2+^:M3^+^ = 3:1 are called hydrotalcite, and quintinite when the ratio M^2+^:M^3+^ = 2:1 [18,19,20]. LDHs have attracted the attention of the scientific community due to their unique properties such as anion exchange ability, thermal stability and easy production [21,22,23] and the promising applications in different fields of adsorption [24,25,26], sensors [27,28,29,30], biomedicine [31,32,33], polymer additives [34,35,36], among others. LDH host layers promote the thermal and optical stabilization of the exchanged anions. Many thermoplastic materials such as polyethylene (PE) do not contain polar groups in their backbone. Therefore, a homogenous dispersion of clay platelets at a nonmetric level is a challenging aim in this technology. The addition of compatibilizers has been proven to favor the exfoliation of hydrotalcite particles [37]. In this work, we have evaluated the effect on the viscoelastic, mechanical and optical properties of the addition of a polar whitening agent to low density polyethylene. A modified LDH hosting the anionic whitening agent has been prepared. The research compares the effect on the polymeric matrix to the addition of the whitening agent, the modified LDH, and also the addition of a commercial surfactant. A characterization of the LDHs has been performed by means of XRD, TGA and FT–IR. The viscoelastic behavior at different temperatures of the resulting nanocomposite has been performed to evaluate the impact on processing conditions. The mechanical and optical properties have been also determined.

## 2. Materials and Methods

### 2.1. Materials

Low density polyethylene (LDPE) was used as the thermoplastic matrix for the formation of the composites. The sample, containing no additives, was purchased by Repsol Technology Center (Madrid, Spain) with reference “LPDE Alcudia PE003” produced by high pressure autoclave technology.

The layered double hydroxide used in this investigation was provided by Kisuma Chemicals (Veedam, The Netherlands) with reference HT4AU from Kisuma Chemicals. The stoichiometric formula is Mg_4_Al_2_(OH)_12_(CO_3_)·nH_2_O with a particle size of 0.4 μm. Therefore, this synthetic LDH is an analogue of mineral quintinite. The anionic exchange capacity is 2.1 mmol/g of LDH in the case of divalent anions and 4.2 mmol/g for monovalent anions. The divalent salt disodium 2,2′-((1,1′-biphenyl)-4,4′-diyldivinylene)bis(benzenesulfonate) was employed as fluorescent whitening agent, chemical compound that absorb light in the ultraviolet region and re-emission in the blue region. The molecular structure is depicted in Scheme 1a and the CAS number is 27344-41-8. To improve the dispersion of the additives in the thermoplastic matrix, sorbitan monooleate with CAS number 1338-43-8 (tradename SPAN80). The formula of this non-ionic surfactant is depicted in Scheme 1b.

### 2.2. Ion Exchange Reaction

One of the most popular anion-exchangeable compounds are LDHs, presenting the highest affinity to carbonate anions. Therefore, to facilitate the introduction of the whitening agent, a precursor was prepared. The first modification of the hydrotalcite is the exchange of carboxylate anions by chloride anions by means of an anion exchange reaction as it is described by Iyi et al. [38]. The reaction consisted on mixing HCl with an aqueous dispersion of LDH under magnetic agitation in N_2_ inert atmosphere, setting the pH to a value of 4.5. 3 anion exchange capacity (AEC) of NaCl (75 g/100 g LDH) are added to introduce Cl anions in the interlayers. The resulting suspension was filtered and washed with degassed water under a nitrogen atmosphere to remove the excess of HCl and NaCl. Then the precipitate was collected and dried in a vacuum to remove water and a white solid is obtained. This modified LDH was labeled as Cl-LDH. The whitening agent (WA) disodium 2,2′-((1,1′-biphenyl)-4,4′-diyldivinylene)bis(benzenesulphonate) was intercalated in Cl-LDH by ionic exchange reaction replacing the chloride anions. In this step, Cl-LDH is mixed with 1 AEC of WA (2.1 mmol of WA per gram of Cl-LDH) for four hours of stirring at room temperature. This exchange reaction was conducted under nitrogen atmosphere and degassed water to avoid the absorption of CO_3_^2−^. The modified hydrotalcite WA-LDH was filtered and washed with degassed water under nitrogen atmosphere to remove the remaining WA and the displaced Cl^−^. Afterwards, the solid was dried in vacuum.

### 2.3. Preparation of Films

The first step of the preparation of films of neat polyethylene (PE) and with additives is a premixture and predispersion in a compounding extruder Leistritz ZSE 18HP (Leistritz Group, Nuremberg, Germany) with intermeshed co-rotating screws. The extruded material is pelletized. Secondly, the nanocomposite pellets were processed in a cast film extrusion line to obtain 200 microns thick films with a Dr. Collin E20P single screw extruder with an L/D ratio of 25 and screw diameter of 20 mm. The final concentration of WA in the nanocomposite is 0.5 wt.%. The extruder temperature ramp is depicted in Table 1. The die width and the calander width were, respectively, 10 and 22 cm. The screw speed set value was 60 rpm, with a die pressure of 45 bares.

### 2.4. Characterization Techniques

In order to evaluate the modification of the LDHs, several characterization techniques have been used. A Leco CHN628 (Leco Corporation, St. Joseph, MI, USA) analyzer was used to determine the elemental composition of the different samples. The Fourier transform infrared analyses were performed in a MIR–FT–IR Bruker 20 Vertex 70–80 device. The thermogravimetric analysis (TGA) was carried out with Mettler Toledo TGA/SDTA 851e/LF1100 equipment (Mettler-Toledo S.A.E., Barcelona, Spain). The X-ray diffraction patterns were recorded on a Bruker D-8 Advance diffractometer (Bruker Biosciences Española S.A., Madrid, Spain) using a wavelength of 1.542 Å from CuKα, with an angular speed of 120 s^−1^, at room temperature.

The rheological study of the PE and its nanocomposites was performed using an AR-G2 rotational rheometer (TA Instruments, New Castle, DE, USA) of parallel plates at different temperatures, with frequency between 10 and 0.01 Hz under a 1% constant strain value. Tensile tests were carried out with a universal test machine ProLine Z010 from Zwick/Roell using a contact extensometer. These tests were accomplished following two standards [39,40]. The optical performance of the nanocomposites has been evaluated with a Perkin Elmer Lambda 750S spectrophotometer with integrating sphere. The absorbance of the sample was recorded in a range from 200 to 800 nm.

## 3. Results

### 3.1. Characterization of the Modified LDHs with WA

The general formula of LDHs is [M^2+^_1−x_M^3+^_x_(OH)_2_]^x+^(A^n−^)_x/n_·mH_2_O. LDH structures exist for values of x in the range of 0.1–0.5, although in pure LDHs the x values has been found to be 0.20–0.33. According to our X-ray fluorescence (XRF) analysis and elemental analysis results, for Cl-LDH, x is equal to 0.49 and 0.45 for WA-LDH. These values higher than 0.33 can be related to an increase of the number of octahedrons containing Al, producing adjacent Al. Al^3+^ ions in the Brucite-like sheets of LDHs are subject to repulsion of positive charges and the increase in neighboring Al octahedra can leads to the formation of amorphous Al(OH)_3_ not detectable by X-ray measurement [12]. This can be ascribed to the formation of an excess on Al^3+^ generated by the removal of magnesium during the preparation of the precursor Cl-LDH. After the first ion exchange reaction, the formation of MgCl by reaction of Mg(OH)_2_ with HCl may occur, and this water-soluble salt can be removed during the washing with the degassed water. At pH = 4.5, this insoluble amorphous Al(OH)_3_ form a passive layer that avoid further Mg^2+^ leaching [41].

The concentration of the elements of both Cl-LDH and WA-LDH was determined by XRF and CHNS. These results are shown in Table 2.

Therefore, according to the data shown in Table 2, the stechiometric formula of Cl-LDH and WA-LDH are:Mg_0.65_Al_0.35_(OH)_2_(Cl)_0.35_0.68H_2_O
g_0.65_Al_0.35_(OH)_2_(WA)_0.18_0.68H_2_O

In Cl-LDH, the carbon content is reduced to 0.34 wt.% and Cl is 14.53 wt.%. Therefore, the chloride anions have satisfactory replaced the carbonate anions in the original hydrotalcite-like material. After the addition of WA, we can observe a decrease in Cl and an increase in S, ascribed to the presence of WA in the LDH. We have calculated the content in WA in WA-LDH from the concentration of S, with a concentration of approximately 47 wt.% of WA in the modified hydrotalcite, corresponding to 0.09 mol/100 g, the amount expected according to the theoretical stechiometric formula for 0.31 mol/100 g of Mg. That means that only 2% of the whitening agent is loss during the reaction. Additionally, according to the XRF results, the concentration of metallic oxides is 52.8 wt.% in Cl-LDH and 25.41 wt.% in WA-LDH, amount greater than the expected pursuant to the stoichiometric formula, which confirms the formation of Al(OH)_3_.

As it can be seen in Figure 1 and Table 3, the unmodified LDH presents a characteristic absorption peak at 1370 cm^−1^, due to the presence of carbonate ions [42]. However, the modified Cl-LDH spectra do not present this peak. Comparing the spectra of WA-LDH and the whitening agent, the peaks at 1174 cm^−1^, appear in both spectra and can be assigned to the presence of sulfonate ions [43]. Therefore, this can be attributed to the incorporation of the whitening agent to the LDH. Thus, the effective reaction exchange is also proved by these data. In Cl-LDH the characteristic absorption peak at 1370 cm^−1^ ascribed to the presence of carbonate anions is not observed, and the incorporation of the whitening agent is also seen in the sample WA-LDH.

The thermogravimetric study of the LDHs and WA is depicted in Figure 2a under an inert atmosphere and Figure 2b under an oxygen atmosphere. The pristine whitening agent presents a first step of mass loss due to the elimination of water. Secondly, the thermal decomposition of the sample starts at 550 °C. For the LDH modified with Cl, there are four steps of mass loss, until 225 °C, 360 °C, 500 °C, and 800 °C. However, in the case of WA-LDH, the mass loss is shifted to higher temperatures. Cl-LDH shows a first mass loss of approximately 19% until 225 °C, which is ascribed to the absorbed water by the material. There is a mass loss of 18% due to the reversible dehydration of the interlaminar hydroxides and water in the second step from 225 to 360 °C [44,45,46]. After that, the loss of HCl can be seen in the range of 360–500 °C, in agreement with the presence of Cl ions [37]. Finally, up to 800 °C the loss of the hydroxide groups present in the Brucite layers can be observed. When the WA is added to LDH, a similar behavior can be seen. However, the loss of HCl is not observed and there is a decomposition of the WA at 550 °C. TGA of WA-LDH shows different steps of mass loss ascribed to hydrotalcite structure: up to 225 °C: loss of absorbed water; 225–450 °C: loss of interlaminar hydroxides; and 500–800 °C: loss of hydroxide groups present in Brucite-like layers. Other steps present in WA-LDH TGA are characteristics of the decomposition of WA: material loss at 500–600 °C and loss of pyrolytic carbon. A subsequent heating of the sample in oxygen atmosphere shows a loss of mass in the pristine whitening. Due to the combustion of the carbon formed in the previous pyrolysis in inert atmosphere, this loss of mass of pyrolytic carbon can also be observed in WA-LDH. However, this mass loss is not present in the thermogravimetric data of CL-LDH, since WA is the only organic compound present. This finding is in agreement with the presence of WA in WA-LDH (Figure 2b).

The evaluation of the structure of the modified LDHs has been performed by means of XRD. Figure 3 depicts the XRD patterns of Cl-LDH and WA-LDH. In Figure 3, the Cl-LDH d003 peak is located at 7.87 Å in contrast with the results obtained with the original LDH (as it is depicted in Figure A1a in Appendix A). However, when the WA-LDH results are examined, there is an increase of the basal d-spacing of WA-LDH having d-value as 19.41 Å in comparison to Cl-LDH having d-value as 7.89 Å (Figure 3b). This can be ascribed to the presence of larger anions in the interlayers due to the effective exchange of Cl with WA [47]. In order to verify that the whitening agent is placed in the interlayers, we have also performed the XRD characterization of the pristine whitening agent (see Figure A1b in Appendix A). These results show that the WA presents a poorly crystalline structure with an intense diffraction peak at 15.17 Å. Although these diffraction peaks could be attributed to the presence of unreacted WA, the smoothness of the pattern in Figure 3b and the results shown in Table 1 are in agreement with an effective exchange reaction in which the WA is located in the interlayers of the hydrotalcite-like material.

### 3.2. Rheology of LDH-Polyolefin Composites

Rheology is a powerful tool that provides crucial knowledge regarding the interaction between the different phases added to polymeric matrices and the processing of composites [48]. In order to perform the rheological characterization of the samples, amplitude sweep tests were performed in a temperature range from 190 to 250 °C to determine the linear viscoelastic region. In every case, constant values of the elastic modulus (G′) and viscous modulus (G″) were found in a wide range of strain. As an example, we have included Figure A2 in Appendix B as supporting information with the determination of the LVR for the PE nanocomposite with WA-LDH with SPAN. G″ values were higher than G′, and when the temperature is raised a decrease of the elastic and viscous module is observed, as expected. Similar trends were found for all the samples, and a constant value of strain of 1% was set to perform the following frequency sweep tests.

We have investigated the effect of the different additives on the viscoelastic behavior of films of polyethylene. In Figure 4, we have shown the values of G′ (a) and G″ (b) for the different samples at 190 °C, which is the lowest temperature utilized in these measurements. It is interesting to notice that G′ and G″ are slightly affected by the addition of either the whitening agent, or the modified LDH. In the case of the WA, there is a low affinity between the anionic molecules and the non-polar macromolecular chain. Thus, the values of the viscoelastic properties do not change with the presence of this additive. There is a small increase on the viscoelastic properties when WA-LDH is added. This finding might indicate that we are reaching the percolation concentration in the melting state. Therefore, an effective interaction between the LDH and PE happens and the viscoelastic properties undergo this limited increase [49]. On the other hand, a drop of G′ and G″ when SPAN is added to the nanocomposite is observed. Electrostatic repulsive forces appear due to the interaction of the hydrophobic chain of SPAN with PE. Therefore, not only does the surfactant improve the dispersion of the nanoclay in the thermoplastic matrix, but also the mobility of the polymeric chains [50]. Then, the positive effect on G′ and G″ when the modified LDH is added is counteracted by the addition of SPAN.

We have performed a series of oscillatory experiments at different temperatures. G′, G″ and complex viscosity data have been recorded, but only the elastic modulus values are depicted in Figure 5. It can be observed that the general tendency of the samples is a decrease of G′ with increasing temperature, as expected [51]. When temperature is raised, the mobility of the macromolecules is increased and the viscoelastic properties diminish. This behavior is observed in our four different films. If we compare Figure 5c with Figure 5d, we can observe the combined effect of surfactant and temperature on the nanocomposite. In both cases, a thinning effect occurs. Paying attention at 240 °C, the values of G′ for the nanocomposites with and without SPAN are very similar. Therefore, at high temperatures the effect of the surfactant seems to be negligible. One of the most important applications of surfactants in thermoplastic materials is to improve their processability. However, in the case of these new nanocomposites, it seems that the addition of surfactant is only necessary in the case of low processing temperatures.

Oscillatory experiments are known to give useful information for processing conditions of thermoplastic-based materials. However, some extreme conditions cannot be investigated due to equipment restrictions, such as high values of oscillatory frequencies. In order to come up to larger values of frequency, Time-Temperature Superposition curves (TTS) have been developed in this work. Figure 6 shows a superposition of the experimental data and the TTS predictions at 240 °C for neat PE (a) and its nanocomposite with WA-modified LDH and surfactant (b). Both TTS curves have been constructed with the oscillatory experiments at 190, 200, 220, and 240 °C. There is a good agreement between the TTS curve and the experimental data in the case of PE. The nanocomposite shows a perfect superposition for the loss modulus, but the storage modulus presents some scattering of the data at low frequency. This effect is probably due to the similarity of the values of G′ at the different temperatures that can be observed in Figure 5d. This low dependence of G′ with the temperature may overcome to a worse accuracy in the treatment of the data. Nevertheless, we can assess that this methodology can predict the behavior of nanoclay-modified composites with a PE matrix.

### 3.3. Mechanical and Optical Properties of the Composites

In order to evaluate actual applications of these new composites, we have performed a series of mechanical and optical tests. In Figure 7 we have represented the results from the tensile tests performed to the neat PE, PE with the addition of WA, PE with the addition of WA-modified LDH and the sample with PE, WA-LDH, and the surfactant. Longitudinal and vertical tests have been carried out and the yield and tensile strengths and the elongation at break have been determined in longitudinal direction of extrusion and also in the transversal direction. In Figure 7a it can be seen that the addition of the additives decrease the mechanical properties of the new nanocomposites, especially when the surfactant is added. It is noteworthy that, for neat PE, the values of the mechanical properties in both directions of the applied load are fairly the same. Longitudinal yield and break strengths are barely affected by the additives. However, a decrease on the transversal mechanical properties is observed. This can be ascribed to lower anisotropy of the polymer chains after the extrusion. Regarding the values of the elongation at break (Figure 7b), the values when the load is applied in the transversal direction of extrusion are lower than the neat PE, but there is no an apparent trend for the longitudinal series.

The optical behavior of the samples is depicted in Figure 8. Neat PE presents the higher transmission of light in UV–VIS range. However, the addition of the WA reduces the transmission at low wavelengths due to the Ultraviolet A (UVA) blocking activity of the whitening agent (blue line). This optical performance is increased if the whitening agent is included in the LDH and even more enhanced when the surfactant is added to the film. LDPE is a non-polar thermoplastic matrix and ionic species as this WA are questionably dispersed. The decrease of UVA transmission in the sample with LDH modified with WA indicates that the optical activity of the whitening agent is enhanced when it is encapsulated between the layers of LDHs. Moreover, if a surfactant (SPAN) is added to the composite, the UVA light transmission is reduced to less than 10%. This effect of surfactant can be attributed to a better dispersion of the LDH in the polymeric matrix [52]. Although the addition of the dispersant provokes a dramatic drop on the transmission of UVA, the transmission in the visible range presents the same values than neat PE. These special optical properties make this material a perfect candidate for greenhouse applications since they are able to only block the damaging light range for plants.

## 4. Discussion

In this work, we have synthesized new double layered-PE composites. The synthesis of the modified LDH has been made by simple exchange reaction. We have proven by means of FT–IR, X-Ray diffraction and TGA that an anionic whitening agent has been incorporated to the LDH. This new modified LDH presents larger interlayer space due to the presence of the WA. In the rheological tests, we have demonstrated that the WA-LDH composites present higher values of G′ and G″, and therefore, the percolation concentration is almost reached. The addition of a dispersant agent decreases the viscoelastic properties of the composite. However, this effect only appears at low temperatures and is negligible when high temperatures are reached (240 and 250 °C). TTS curves have been elaborated to increase the frequency window of the films. The anisotropy of the films provoke that the surfactant has a negative impact on the transversal mechanical properties of the resulting composites, but with small effect on the longitudinal yield and break strengths. Finally, the addition of WA decreases the transmission of the UVA light but not in the visible range, and the best results are found for the formulation with encapsulated WA and surfactant. Therefore, the addition of surfactant not only enhances the processing of the nanocomposites but their optical properties.

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
