# Peer review of "Synthesis and Characterization of New Layered Double Hydroxide-Polyolefin Film Nanocomposites with Special Optical Properties"

_materials, 2019, doi:10.3390/ma12213580_

Round 1

Reviewer 1 Report

Review report for the paper «Synthesis and Characterization of New Layered Double Hydroxide-Polyolefin Film Nanocomposites with Special Optical Properties» prepared by Monzó et al., 2019.

Lines 37-39 classical reference regarding crystal chemistry of LDHs should be used instead of references 7, 8. “Layered double hydroxides (LDH) are anionic clays represented by the general formula of 37 [M2+1-xM3+x(OH)2]x+(An-)x/n·mH2O, where M2+ and M3+ are divalent and trivalent metallic cations and 38 An- depicts the anions which can be present in the hydrated interlayer galleries of the clay.” Was it discovered in 2012-2017 by Chen et al. and Cuncha et al? No! See at least Rives et al., 2001 https://books.google.ru/books/about/Layered_Double_Hydroxides.html?id=U5c5zzqjthgC&redir_esc=y and Evans and Slade 2005 https://www.springer.com/gp/book/9783540282792.  

Lines 42-43 “These anions are weakly bound to the 42 structure and ion exchange reactions are suitable for the modification of LDHs [12]”. It has been known prior Zhao et al., 2003 and the work of  Zhao et al. doesn`t discuss theoretical background of the strengths of layer-interlayer bonding sine “A new method of synthesizing hydrotalcite-like layered double hydroxides (LDHs) of the type [Mg1-xAlx(OH)2]x+(CO32)x/2·yH2O (x = 1.7−3.3) is reported”.

Lines 43-44 This is not true. Two Mg-Al-CO3 minerals are known: hydrotalcite Mg6Al2(CO3)(OH)16 · 4H2O and quintinite Mg4Al2(OH)12(CO3) · 3H2O which is principal difference since 3:1 and 2:1 LDHs exhibit different properties. See nomenclature https://pubs.geoscienceworld.org/minmag/article/76/5/1289/85510/nomenclature-of-the-hydrotalcite-supergroup and report on crystal chemistry of hydrotalcite discussing prolonged confusion between two minerals https://www.cambridge.org/core/journals/mineralogical-magazine/article/crystal-chemistry-of-natural-layered-double-hydroxides-5-singlecrystal-structure-refinement-of-hydrotalcite-mg6al2oh16co3h2o4/E15DF425809CDB557073AAF92CBF9157.

Lines 44-45 It is not correct to postulate that “In the recent years, LDH have attracted the attention of the scientific community due to their unique properties such as anion exchange ability”. It should be either about recent advances (which are not anion-exchange properties that are known for decades) or about that properties that have been known for a while. Citing of review articles on mentioned properties is requested.

Line 48 Reference 22 indicated 2 times in a row.

Line 68 You have been working with synthetic analogue of the mineral quintinite.

Line 72 Double dot.

Section 2.1 It is worth mentioning the seize and the charge of whitening agent.

Line 86 interlaryers.

Line 100 the Concentration of the elements of Cl-LDH and WA-LDH given in Table 1 should be recalculated to chemical formula. Chemical formulas should be given as Table footnote or inserted in Table 1 or given in that section.

Line 114 “Fourier Transform Infra Red” (should be infrared).

Line 118 CuKa source.

Lines 129-138 Please state more clearly: 1) why only Mg was removed and Al remained and 2) what form of Al is responsible for its high content (do authors think that Al(OH)3 formed? “producing adjacent Al” it not clear.

Line 137 Just a note for authors for their future work that LDH corresponding to chemical composition Mg0.65Al0.35(OH)2(Cl)0.350.68H2O is known as mineral chlormagaluminite.

Line 140 in the original synthetic quintinite or “hydrotalcite” or hydrotalcite-like material.

Line 143 synthetic quintinite or “hydrotalcite” or hydrotalcite-like material instead of hydrotalcite (terminology matters).

Line 151 remove y.

Line 148-151 It is mandatory to provide a Table with wavenumbers determined for different spectra, their assignment and comparison with reference species (taken from literature data) e.g.

Table X. Infrared bands observed for unmodified LDH, Cl-LDH and WA-LDH compared to literature data

Wavenumbers observed for

So-called unmodified LDH (cm-1)

Wavenumbers observed for natural (quintinite) species or well-characterized synthetic analogue

Wavenumbers observed for

Cl-LDH

Wavenumbers observed for natural (chlormag-aluminite) species or well-characterized synthetic analogue

Wavenumbers observed for WA-LDH

Wavenumbers observed for whitening agent

Band Assignment

3453

3444

3456

3467

3567

3546

3546

Lines 155-169 Please develop description of mass loses for WA-LDH what mass loss occurred in what temperature range? Does it correspond to theoretical formula of WA-LDH? Mandatory change.

The first mass loss of Cl-LDH should be ascribed to the loss of interlayer water (~11 or more %) and absorbed water. Read more literature regarding that.

“There is a mass loss of 18 % due to the reversible dehydration of the interlaminar hydroxides and water in the second step from 162 225 to 360 ºC. 163” It should be assigned to the loss of OH groups (reversible).

Line 160 “The presence of Cl in Cl-LDH is also confirmed.” How? It cannot be stated like this in Results.

Line 170 I didn`t get from the text why Cl-LDH was not losing mass in Oxygen atmosphere.

Line 174 Figure 3 depicts the XRD patterns of Cl-LDH and WA-LDH.

Line 174 “In the case of Cl-LDH the d003 peak is shifted to 7.87 Å showing a characteristic pattern of LDHs as it is seen in Figure 3a 175”. d003 => d003 (Å). Why shifted (shifted in comparison to what)? It is simply located there. Compare obtained d-value with other Cl-LDH. Does this value correspond to reference species?

Line 176 “there is an increase of the basal d-spacing and a shift of d003” The shift of d003 relative to what? D003 is enough sine d006 and d009 is a duplication. D003 of WA-LDH should be given in the text like ‘there is an increase of the basal d-spacing of WA-LDH having d-value as 19.41 Å in comparison to Cl-LDH having d-value as 7.89 Å’.

Line 184 replace hydrotalcite (see above).

Line 188 What for authors report rheological characterization. Is it important for technological application? Please explain at the beginning of paragraph. Why exactly 190-250 C range was chosen?

Line 190 decrypt what is G’ and G’’.

Decrypt abbreviation.

References The literature list should be significantly extended.

Author Response

Comment 1.

Lines 37-39 classical reference regarding crystal chemistry of LDHs should be used instead of references 7, 8. “Layered double hydroxides (LDH) are anionic clays represented by the general formula of 37 [M2+1-xM3+x(OH)2]x+(An-)x/n·mH2O, where M2+ and M3+ are divalent and trivalent metallic cations and 38 An- depicts the anions which can be present in the hydrated interlayer galleries of the clay.” Was it discovered in 2012-2017 by Chen et al. and Cuncha et al? No! See at least Rives et al., 2001 https://books.google.ru/books/about/Layered_Double_Hydroxides.html?id=U5c5zzqjthgC&redir_esc=y and Evans and Slade 2005 https://www.springer.com/gp/book/9783540282792.  

Answer:

The reviewer is right. We have added the following references:

Jones, W., Chidwe, M.; Pillared Layered Structures – Current Trends and Applications. 1990 I.V. Mithecll, 7’’Ed. p. 67. Elsevier Applied Science, London and New York. Rives, V. Layered Double Hydroxides: Present and Future 2001, Nova Publishers. Ma, S.; Fan, C.; Du, L.; Huang, G.; Yang, X.; Tang, W.; Makita Y.; Ooi, K.; Synthesis, Anion Exchange, and Delamination of Co−Al Layered Double Hydroxide:  Assembly of the Exfoliated Nanosheet/Polyanion Composite Films and Magneto-Optical Studies Chem. Mater., 2009, 21, 3602–3610.

Comment 2.

Lines 42-43 “These anions are weakly bound to the 42 structure and ion exchange reactions are suitable for the modification of LDHs [12]”. It has been known prior Zhao et al., 2003 and the work of  Zhao et al. doesn`t discuss theoretical background of the strengths of layer-interlayer bonding sine “A new method of synthesizing hydrotalcite-like layered double hydroxides (LDHs) of the type [Mg1-xAlx(OH)2]x+(CO32)x/2·yH2O (x = 1.7−3.3) is reported”

Answer:

We have added the following references about the ion exchange reactions of LDHs:

Miyata, S.; Kumura, T.; SYNTHESIS OF NEW HYDROTALCITE-LIKE COMPOUNDS AND THEIR PHYSICO-CHEMICAL PROPERTIES, Chem. Lett. 1973, 2, 843-848. Meyn, M.; Beneke, K.; Lagaly, G. Anion-Exchange Reactions of Layered Double Hydroxides. Inorg. Chem. 1990, 29, 5201−5207.

Comment 3.

Lines 43-44 This is not true. Two Mg-Al-CO3 minerals are known: hydrotalcite Mg6Al2(CO3)(OH)16 · 4H2O and quintinite Mg4Al2(OH)12(CO3) · 3H2O which is principal difference since 3:1 and 2:1 LDHs exhibit different properties. See nomenclature https://pubs.geoscienceworld.org/minmag/article/76/5/1289/85510/nomenclature-of-the-hydrotalcite-supergroup and report on crystal chemistry of hydrotalcite discussing prolonged confusion between two minerals https://www.cambridge.org/core/journals/mineralogical-magazine/article/crystal-chemistry-of-natural-layered-double-hydroxides-5-singlecrystal-structure-refinement-of-hydrotalcite-mg6al2oh16co3h2o4/E15DF425809CDB557073AAF92CBF9157

Answer:

We agree with the referee and we have rephrased the sentences as follows “LDHs composed of Mg and Al hydroxyl sheets with carbonate ions as counter anions and M2+:M3+ = 3:1 are called hydrotalcite, and quintinite  when ratio M2+:M3+ = 2:1 [18,19]” and added the suggested references as new references 18 and 19.

Comment 4:

Lines 44-45 It is not correct to postulate that “In the recent years, LDH have attracted the attention of the scientific community due to their unique properties such as anion exchange ability”. It should be either about recent advances (which are not anion-exchange properties that are known for decades) or about that properties that have been known for a while. Citing of review articles on mentioned properties is requested.

Answer:

We have removed “in the recent years” and have added the following references:

Daud, M.; Hai, A.; Banat, F.; Wazir, M.B.; Habib, M.; Bharath, G.; Al-Harthi, M.A. A Review on the recent advances, challenges and future aspect of layered double hydroxides (LDH) - Containing hybrids as promising adsorbents for dyes removal. J. Mol. Liq. 2019, 288, 110989. Baig, N.; Sajid, M. Applications of layered double hydroxides based electrochemical sensors for determination of environmental pollutants: A review. Trends Environ. Anal. 2017, 16, 1-15. Asif, M.; Aziz, A.; Azeem, M.; Wang, Z.Y.; Ashraf, G.; Xiao, F.; Chen, X.D.; Liu, H.F. A review on electrochemical biosensing platform based on layered double hydroxides for small molecule biomarkers determination. Adv. Colloid Int. Sci. 2018, 262, 21-38. Yan, L.; Gonca, S.; Zhu, G.Y.; Zhang, W.J.; Chen, X.F. Layered double hydroxide nanostructures and nanocomposites for biomedical applications. J. Mater. Chem. B. 2019, 7, 5583-5601. Jia, L.; Ma, J.Z.; Gao, D.G.; Lv, B. Layered Double Hydroxides/Polymer Nanocomposites. Prog. Chem. 2018, 30, 295-303.

Comment 5:

Line 48 Reference 22 indicated 2 times in a row.

Answer 5: Fixed.

Comment 6:

Line 68 You have been working with synthetic analogue of the mineral quintinite.

Answer:

True. We have added in line 70 the following sentence to make this statement clear: “Therefore, this synthetic LDH is an analogue of mineral quintinite.”

Comment 7:

Line 72 Double dot.

Answer: fixed.

Comment 8:

Section 2.1 It is worth mentioning the seize and the charge of whitening agent.

Answer:

We have rephrased the sentence as “The divalent salt disodium 2,2’-([1,1’-biphenyl]-4,4’-diyldivinylene)bis(benzenesulphonate) was employed as fluorescent whitening agent, chemical compound that absorb light in the ultraviolet region and re-emission in the blue region”. Regarding the size, we do not have any technique available in our facilities to measure the size of such small molecule. Dynamic Light Scattering could be an option, but since it is a very polar molecule, the hydration corona around the molecule will give an apparent hydrodynamic radius much larger than the actual size of the molecule.

Comment 9:

Line 86 interlaryers.

Answer: Fixed.

Comment 10:

Line 100 the Concentration of the elements of Cl-LDH and WA-LDH given in Table 1 should be recalculated to chemical formula. Chemical formulas should be given as Table footnote or inserted in Table 1 or given in that section.

Answer:

We have reorganized this part of the paper, and Table 1 is now shown in the Results section as Table 2 as follows:

“The concentration of the elements of both Cl-LDH and WA-LDH was determined by XRF and CHNS. These results are shown in Table 2.

Table 2. Concentration in wt% and mol of the elements of Cl-LDH and WA-LDH determined by XRF and CHNS.

Concentration (wt.%)

Concentration (mol/g)

Elements

Cl-LDH

WA-LDH

Cl-LDH

WA-LDH

Na

2.03

0.15

0.09

0.006

Mg

13.55

7.45

0.56

0.31

Al

14.63

6.80

0.54

0.25

S

0.058

5.88

0.002

0.18

C

0.34

30.80

0.03

2.57

Cl

14.73

0.024

0.41

0.0007

Therefore, according to the data shown in Table 2, the stechiometric formula of Cl-LDH and WA-LDH are:

Mg0.65Al0.35(OH)2(Cl)0.350.68H2O

Mg0.65Al0.35(OH)2(WA)0.180.68H2O”

Comment 11:

Line 114 “Fourier Transform Infra Red” (should be infrared).

Answer: Fixed

Coment 12:

Line 118 CuKa source.

Answer: Fixed

Comment 12:

Lines 129-138 Please state more clearly: 1) why only Mg was removed and Al remained and 2) what form of Al is responsible for its high content (do authors think that Al(OH)3 formed? “producing adjacent Al” it not clear.

Answer:

This is a very good question  and we have found that due to the pH values of 4.5 used in this reaction, the formation of a layer of aluminium hydroxide prevents from  magnesium leaching, as it is explained by Jobagy et al. (reference 39). We have added the following discussion: “The general formula of LDHs is [M2+1-xM3+x(OH)2]x+(An-)x/n·mH2O. LDH structures exist for values of x in the range of 0.1-0.5, although in pure LDHs the x values has been found to be 0.20-0.33. According to our XRF and CHNS results, for Cl-LDH, x is equal to 0.49 and 0.45 for WA-LDH. These values higher than 0.33 can be related to an increase of the number of octahedrons containing Al, producing adjacent Al. Al3+ ions in the Brucite-like sheets of LDHs are subject to repulsion of positive charges and the increase in neighboring Al octahedra can leads to the formation of amorphous Al(OH)3 not detectable by X-ray measurement [12]. This can be ascribed to the formation of an excess on Al3+ generated by the removal of magnesium during the preparation of the precursor Cl-LDH. After the first ion exchange reaction, the formation of MgCl by reaction of Mg(OH)2 with HCl may occur, and this water-soluble salt can be removed during the washing with the degassed water. At pH = 4.5, this insoluble amorphous Al(OH)3 form a passive layer that avoid further Mg2+ leaching [39]. “

Comment 13:

Line 137 Just a note for authors for their future work that LDH corresponding to chemical composition Mg0.65Al0.35(OH)2(Cl)0.350.68H2O is known as mineral chlormagaluminite.

Answer: Thank you for the suggestion.

Comment 14:

Line 140 in the original synthetic quintinite or “hydrotalcite” or hydrotalcite-like material.

Answer: We have replaced “hydrotalcite” by “hydrotalcite-like material”.

Comment 15:

Line 143 synthetic quintinite or “hydrotalcite” or hydrotalcite-like material instead of hydrotalcite (terminology matters).

Answer: We totally agree with the referee and we have replaced hydrotalcite by “modified synthetic quintinite”.

Comment 16:

Line 151 remove y.

Answer: Fixed. We have replaced “y” by “and”.

Comment 17:

Line 148-151 It is mandatory to provide a Table with wavenumbers determined for different spectra, their assignment and comparison with reference species (taken from literature data) e.g.

Table X. Infrared bands observed for unmodified LDH, Cl-LDH and WA-LDH compared to literature data

Wavenumbers observed for

So-called unmodified LDH (cm-1)

Wavenumbers observed for natural (quintinite) species or well-characterized synthetic analogue

Wavenumbers observed for

Cl-LDH

Wavenumbers observed for natural (chlormag-aluminite) species or well-characterized synthetic analogue

Wavenumbers observed for WA-LDH

Wavenumbers observed for whitening agent

Band Assignment

3453

3444

3456

3467

3567

3546

3546

Answer: We agree with the referee and the data are better presented with the following table.

Table 3. Infrared bands observed for unmodified LDH, Cl-LDH and WA-LDH compared to literature data.

Wavenumber (cm-1)

Originial LDH

Quintinite[40]

Cl-LDH

WA-LDH

WA

Band assignment

1370

1350

-

-

-

CO32− ν3 antisymmetric stretching

-

-

-

1174

1180

SO3- ν3 antisymmetric stretching

3415

3388

3395

3467

-

OH- stretching vibration

Comment 18:

Lines 155-169 Please develop description of mass loses for WA-LDH what mass loss occurred in what temperature range? Does it correspond to theoretical formula of WA-LDH? Mandatory change.

The first mass loss of Cl-LDH should be ascribed to the loss of interlayer water (~11 or more %) and absorbed water. Read more literature regarding that.

“There is a mass loss of 18 % due to the reversible dehydration of the interlaminar hydroxides and water in the second step from 162 225 to 360 ºC. 163” It should be assigned to the loss of OH groups (reversible).

Answer:

Regarding the WA-LDH thermogravimetric analysis we have added the following description:

“When the WA is added to LDH, a similar behavior can be seen. However, the loss of HCl is not observed and there is a decomposition of the WA at 550 ºC. TGA of WA-LDH shows different steps of mass loss ascribed to hydrotalcite structure; till 225ºC loss of absorbed, 225-450 ºC loss interlaminar hydroxides and 500-800 ºC loss of hydroxide groups present in Brucite-like layers. Other steps presents in WA-LDH TGA are characteristics of the decomposition of WA; material loss at 500-600 ºC and loss of pyrolytic carbon. A subsequent heating of the sample in oxygen atmosphere shows a loss of mass in the Pristine Whitening. Due to the combustion of the carbon formed in the previous pyrolysis in inert atmosphere, this loss of mass of pyrolytic carbon can also be observed in WA-LDH. However, this mass loss is not present in the thermogravimetic data of CL-LDH, since WA is the only organic compound present. This finding is in agreement with the presence of WA in WA-LDH (Figure 2b).”

However, TGA analysis of WA-LDH cannot confirm its theorical formula, the interaction of the hydrotalcite components such as metals and hydroxides with WA during pyrolysis can generate different degradation subproducts and ashes, comparing to those obtained in the pristine WA pyrolysis, deeper studies about this subproducts would be needed.

Regarding the mass loss in Cl-LDH, this is the only comment that we disagree with the reviewer. We have checked the literature and there are other works with similar interpretation of our data (see references 42, 43 and 44 ).

Comment 19:

Line 160 “The presence of Cl in Cl-LDH is also confirmed.” How? It cannot be stated like this in Results.

Answer:  The referee is right. We have rephrased the sentence as follows: “After that, the loss of HCl can be seen in the range of 360 to 500 ºC, in agreement with the presence of Cl ions [37].”

Comment 20

Line 170 I didn`t get from the text why Cl-LDH was not losing mass in Oxygen atmosphere.

Answer:

Sorry for the unclear explanation. We have rephrased the very long sentence as follows: “A subsequent heating of the sample in oxygen atmosphere shows a loss of mass in the Pristine Whitening. Due to the combustion of the carbon formed in the previous pyrolysis in inert atmosphere, this loss of mass of pyrolytic carbon can also be observed in WA-LDH. However, this mass loss is not present in the thermogravimetic data of CL-LDH, since WA is the only organic compound present. This finding is in agreement with the presence of WA in WA-LDH (Figure 2b).“

Comment 21:

Line 174 Figure 3 depicts the XRD patterns of Cl-LDH and WA-LDH.

Answer: Fixed

Comment 22:

Line 174 “In the case of Cl-LDH the d003 peak is shifted to 7.87 Å showing a characteristic pattern of LDHs as it is seen in Figure 3a 175”. d003 => d003 (Å). Why shifted (shifted in comparison to what)? It is simply located there. Compare obtained d-value with other Cl-LDH. Does this value correspond to reference species?

Answer:

We have rephrased the sentence as follows: “The evaluation of the structure of the modified LDHs has been performed by means of XRD. Figure 3 depicts the XRD patterns of Cl-LDH and WA-LDH. In Figure 3, the Cl-LDH d003 peak is located at 7.87 Å in contrast with the results obtained with the original LDH (as it is depicted in Figure A1(a)). However, when the WA-LDH results are examined, there is an increase of the basal d-spacing of WA-LDH having d-value as 19.41 Å in comparison to Cl-LDH having d-value as 7.89 Å (Figure 3b). This can be ascribed to the presence of larger anions in the interlayers due to the effective exchange of Cl with WA [45].”

Comment 23:

Line 176 “there is an increase of the basal d-spacing and a shift of d003” The shift of d003 relative to what? D003 is enough sine d006 and d009 is a duplication. D003 of WA-LDH should be given in the text like ‘there is an increase of the basal d-spacing of WA-LDH having d-value as 19.41 Å in comparison to Cl-LDH having d-value as 7.89 Å’.

Answer: We have rephrased the sentence as the referee has suggested.

Comment 24:

Line 184 replace hydrotalcite (see above).

Answer: Fixed

Comment 25:

Line 188 What for authors report rheological characterization. Is it important for technological application? Please explain at the beginning of paragraph. Why exactly 190-250 C range was chosen?

Answer: The viscoelastic properties determine the conditions of processing and the mechanical behavior of thin films. Therefore, the rheological behavior of the sample is directly related to the processing of the composites. The inclusion of LDHs may change the flowing behavior of the melt to obtain films. Since we study simultaneously the effect of stress and temperature, we give relevant information about the processing conditions. PE is usually extruded at temperatures about 200 ºC, and we have explored the possibility to increase the temperature.

We have added the following sentence: “Rheology is a powerful tool that provides crucial knowledge regarding the interaction between the different phases added to polymeric matrices and the processing of composites.”

Comment 26:

Line 190 decrypt what is G’ and G’’.

Answer:  We have modified the text as follows:  “In every case, constant values of elastic modulus (G’) and viscous modulus (G’’) were found in a wide range of strain”

Comment 27:

References The literature list should be significantly extended.

Answer: We have added  23 more references.

Reviewer 2 Report

This manuscript has several issues not clear and can be revised as follows.

1.  The special optical properties emphasized in the title of this manuscript is not clearly shown in the abstract.

2.  The "LTPE" and "whitening agent" should have to be define in the abstract.

3. Why did the addition of WA decrease the transmission of the UVA light but not in the visible range? Please give the mechanism in detail, not only explain by "a better dispersion of the LDH in
269 the polymeric matrix [27]".

4. This special optical properties make this material a perfect candidate for greenhouse applications. Please give the reason that the effect resulting from the special optical properties.

5. In Fig. 7(b), why did the sample "PE+WA-LDH+surfactant" show the highest Elongation at break (%)?

6. The addition of a dispersant agent decreases the viscoelastic properties of the composite,but this effect is negligible when high temperatures are reached 240ºC. Can this effect be reversible? 

7. The environment reliability for the proposed new double layered hydroxides with LDPE thermoplastic matrix and whitening agent is also very important for applications. Please show the related life data if possible. 

Author Response

Comments  1 and 2:

The special optical properties emphasized in the title of this manuscript is not clearly shown in the abstract. The "LTPE" and "whitening agent" should have to be define in the abstract.

Answer:

We have modified the abstract as follows:

In this study, we have synthesized new double layered hydroxides to be incorporated to low density polyethylene thermoplastic matrix. These new composites present promising applications as materials to build greenhouses due to the enhancement of their optical properties. A characterization of the modified nanoclay has been performed by means of XRD, TGA and FT-IR. We have prepared a series of polyolefin-based films to evaluate the effect of the addition of a whitening agent (disodium 2,2’-([1,1’-biphenyl]-4,4’-diyldivinylene)bis(benzenesulfonate)), the modified hydrotalcite-like material and a commercial dispersant. The rheological and mechanical characterization of the films have proved that the inclusion of the modified LDHs do not substantially affect the processing and mechanical performance of the material. On the other hand, optical properties of the nanocomposites are improved by reducing the transmission in the UVA region.

Comment 3 and 4:

Why did the addition of WA decrease the transmission of the UVA light but not in the visible range? Please give the mechanism in detail, not only explain by "a better dispersion of the LDH in the polymeric matrix [27]". This special optical properties make this material a perfect candidate for greenhouse applications. Please give the reason that the effect resulting from the special optical properties.

Answer:

One of the main problems of plastic with greenhouses applications is that PE has a large transmission in the UVA range. This kind of light is harmful to plants and the screening of these wavelengths is desirable, as far as the visible range light is transmitted (necessary for the photosynthesis of the plants). Whitening agents are known to block UVA light but transmit visible light. However, they are hardly dispersed into polymeric matrices because of their polar nature. In this work, we have found that large ionic WAs can be encapsulated in LDHs and the optical performance is improved.

We have rephrased the last paragraph as follows:

“The optical behavior of the samples is depicted in Figure 8. Neat PE presents the higher transmission of light in UV-visible range. However, the addition of the WA reduces the transmission at low wavelengths due to the UVA blocking activity of the whitening agent (blue line). This optical performance is increased if the whitening agent is included in the LDH and even more enhaced when the surfactant is added to the film. LDPE is a non-polar thermoplastic matrix and ionic species as this WA are questionably dispersed. The decrease of UVA transmission in the sample with LDH modified with WA indicates that the optical activity of the whitening agent is enhanced when it is encapsulated between the layers of LDHs. Moreover, if a surfactant (SPAN) is added to the composite, the UVA light transmission is reduced to less than 10 %. This effect of surfactant can be attributed to a better dispersion of the LDH in the polymeric matrix [43]. Although the addition of the dispersant provokes a dramatic drop on the transimission of UVA, the transmission in the visible range presents the same values than neat PE. These special optical properties make this material a perfect candidate for greenhouse applications since they are able to only block the damaging light range for plants.”

Comment 5:

 In Fig. 7(b), why did the sample "PE+WA-LDH+surfactant" show the highest Elongation at break (%)?

Answer:

The samples do not follow a clear tendency, but it is possible that the surfactant has a lubricating effect and the polymer chains of PE have a higher mobility. Then, the film can be more easily deformed probably because these chains are oriented in the flow direction. But we do not have any other property to prove this statement.

Comment 6:

 The addition of a dispersant agent decreases the viscoelastic properties of the composite,but this effect is negligible when high temperatures are reached 240ºC. Can this effect be reversible?

Answer:

At this high temperature the mobility of the chains in the melt is that high that there is no effect if the surfactant is added. In principle, all the interactions between the surfactant and the polymer are driven by reversible forces, bur at this temperature some degradation of the sample may occur if the experiment lasts too long. Therefore, reversibility has not been checked, but it can be expected.

Comment 7:

 The environment reliability for the proposed new double layered hydroxides with LDPE thermoplastic matrix and whitening agent is also very important for applications. Please show the related life data if possible.

Answer:

This is a very good question. We have also thought about these aspects and we are currently running new investigations about the durability of these nanocomposites.

Reviewer 3 Report

The publication of the paper “Synthesis and Characterization of New Layered Double Hydroxide-Polyolefin Film Nanocomposites with Special Optical Properties” is recommended after minor revisions:

Page 3 Line 83 The reference is incorrect, it is Iyi et al. [25] not Iyi et al. [24].

Page 3 Line 84. At which solution pH it has been performed the substitution of CO32- by Cl to obtain the Cl-LDH? This is a fundamental parameter which should be specified because it would be useful to understand the changes in the LDH stoichiometry.

Page 3 Table 1. Please report the correct units of concentration in table 1, mol/g, mol/Kg ...?

In the "results" section If possible it would be useful to report (e.g. in figure 3) the XRD pattern of original carbonate LDH in order to have a comparison with the other patterns. In fact the replacement of CO32- by Cl should be confirmed by the shift of LDH basal reflections; moreover, if the molar ratio Mg2+/Al3+ changes from 2 to 3, as a consequence of the loss of Mg and the formation of Al(OH)3, it should be visible through the shift of the (110) peak (at about 60° 2θ) which is used to define the LDH structural parameter a (see for example Cavani et al. [9] and Vaccari [10]

Author Response

Comment 1:

Page 3 Line 83 The reference is incorrect, it is Iyi et al. [25] not Iyi et al. [24].

Answer: Fixed

Comment 2:

Page 3 Line 84. At which solution pH it has been performed the substitution of CO32- by Cl− to obtain the Cl-LDH? This is a fundamental parameter which should be specified because it would be useful to understand the changes in the LDH stoichiometry.

Answer: The referee is completely right and we apologize for this mistake. The pH reaction was set to 4.5, and we have added this data as follows: “The reaction consisted on mixing HCl with an aqueous dispersion of LDH under magnetic agitation in N2 inert atmosphere, setting the pH to a value of 4.5.”

Comment 3:

Page 3 Table 1. Please report the correct units of concentration in table 1, mol/g, mol/Kg …?

Answer:

Once again, we apologize for this mistake. It is mol/g and it has been changed in the table, which now is Table 2.

Comment 4:

In the “results” section If possible it would be useful to report (e.g. in figure 3) the XRD pattern of original carbonate LDH in order to have a comparison with the other patterns. In fact the replacement of CO32- by Cl− should be confirmed by the shift of LDH basal reflections; moreover, if the molar ratio Mg2+/Al3+ changes from 2 to 3, as a consequence of the loss of Mg and the formation of Al(OH)3, it should be visible through the shift of the (110) peak (at about 60° 2θ) which is used to define the LDH structural parameter a (see for example Cavani et al. [9] and Vaccari [10]

Answer:

The XRD pattern of the original LDH is not that different from Cl-LDH. It is true that there is a small shift of the basal reflections, but it is not conclusive. We have added the figure as an appendix.

Round 2

Reviewer 2 Report

The reviewer's comments have been explained or answered in the response letter, also in the revised manuscript. This manuscript can now be accepted in the present form.